

# Efficient context-aware computing: a systematic model for dynamic working memory updates in context-aware computing

Mumtaz Ali[1], Muhammad Arshad[1], Ijaz Uddin[1], Muhammad Binsawad[2], Abdullah Bin Sawad[3] and Osama Sohaib[4,5]

[1] Department of Computer Science, City University of Science and Information Technology, Peshawar, Pakistan
[2] Department of Information Systems, Faculty of Computing and Information Technology, King Abdulaziz University, Jeddah, Saudi Arabia
[3] Department of Computer and Information Technology, The Applied College, King Abdulaziz University, Jeddah, Saudi Arabia
[4] School of Computer Science, University of Technology Sydney, Sydney, Australia
[5] School of Business, American University of Ras al Khaimah, Ras al Khaimah, United Arab Emirates

Corresponding authors
Mumtaz Ali, mumtazali@cusit.edu.pk
Osama Sohaib,
Osama.Sohaib@uts.edu.au

## ABSTRACT

The expanding computer landscape leads us toward ubiquitous computing, in which smart gadgets seamlessly provide intelligent services anytime, anywhere. Smartphones and other smart devices with multiple sensors are at the vanguard of this paradigm, enabling context-aware computing. Similar setups are also known as smart spaces. Context-aware systems, primarily deployed on mobile and other resource-constrained wearable devices, use a variety of implementation approaches. Rule-based reasoning, noted for its simplicity, is based on a collection of assertions in working memory and a set of rules that regulate decision-making. However, controlling working memory capacity efficiently is a key challenge, particularly in the context of resource-constrained systems. The paper's main focus lies in addressing the dynamic working memory challenge in memory-constrained devices by introducing a systematic method for content removal. The initiative intends to improve the creation of intelligent systems for resource-constrained devices, optimize memory utilization, and enhance context-aware computing.

## INTRODUCTION

The latest computing trends drive toward ubiquitous computing, where seamlessly integrated gadgets, leveraging embedded or portable devices, deliver intelligent services to assist individuals anytime and anywhere. This paradigm empowers users to communicate and exchange information through smart gadgets, enhancing convenience, safety, and overall well-being, albeit with device dependency. With their inherent mobility, smartphones align perfectly with this vision, benefiting from a wide range of

communication channels and incorporating sensors like GPS, proximity, shake sensors, and accelerometers (*Pei et al., 2010*). These sensors enhance users' daily activities by enabling access to many smart applications that provide contextual information about the user and their environment, thus contributing to a context-aware computing paradigm.

Context awareness, initially centered around the positioning of objects and people in early works on Pervasive and Ubiquitous Computing (*Schilit & Theimer, 1994*), has expanded in recent years to encompass additional dimensions such as social and physical aspects of entities, as well as user activities and behavior towards their surroundings (*Rakib & Uddin, 2019*; *Dourish, 2004*; *Sarker, 2019*; *Brown et al., 2022*). These sensors offer valuable user information and facilitate seamless communication among users, devices, and applications (*Raento et al., 2005*). Smartphones, self-contained entities equipped with sensors, code, and applications, can qualify as agents in agent-based computing, functioning autonomously in their respective domains. Consequently, using agent-based systems and smartphones holds immense potential for developing sophisticated context-aware systems.

Considering these factors, designing and developing context-aware smart systems that can be deployed universally becomes an ideal approach. Leveraging existing data, these systems can intelligently infer outputs or adapt to new contexts in real time. Developing intelligent context-aware systems that can think and make decisions on behalf of users or other devices using a rule-based system (RBS) emerges as a viable solution. RBS utilizes rules to steer the system in the desired direction. The ultimate goal is to integrate such intelligent systems into small gadgets, thereby significantly enhancing the quality of life, especially for individuals with disabilities, who may require minimal input or effort.

The present study continues the work presented in *Ali et al. (2024)*. A memory calculation model calculates the required working memory for application(s). This model calculates working memory requirements very efficiently; however, the model still lacks the removal/withdrawal of preference sets from the memory that are no longer in use. Due to this, the memory reaches its maximum, and then there is no systematic method for removing rules from the memory. Currently, the benchmark technique (*Rakib & Uddin, 2019*) eliminates rules randomly from the memory once it reaches its limits, which leads to different issues discussed in detail in *Ali et al. (2024)*, especially the removal of critical context/rules. With the introduction of our previous work and after the incorporation of the proposed technique, the model optimization process reached its conclusion, and now we can claim to have an efficient solution.

The rest of this article is organized as follows. 'Related Work' provides a review of relevant literature. 'Working Memory Size Estimation' presents the working memory size estimation model and its algorithm. 'Working Memory Update Methodologies' provides a detailed discussion of the working memory updating model. 'Scenarios' provides an insight into the scenarios and their results. Lastly, 'Conclusion' summarizes the key findings of the article.

## RELATED WORK

The related work primarily focuses on two key subsections: "Context-Aware Systems," which provides a detailed discussion of the topic, and "Preferences," which highlights the significant role played by preferences in the contemporary era, enhancing system efficiency.

### Context-aware systems

Context awareness is becoming increasingly crucial in pervasive and ubiquitous computing. As early works *Schilit & Theimer (1994)* illustrate, context awareness was first limited to the location of objects and people. Recent advancements, however, have broadened the concept of context to include additional components, such as the social and physical features of entities and user behaviors (*Rakib & Uddin, 2019*; *Dourish, 2004*; *Sarker, 2019*). To comprehend the many characteristics of context, researchers conducted rigorous evaluations. Studies have examined user identity, social situations, location information, temporal data, and the surrounding environment, including objects and individuals (*Schilit & Theimer, 1994*; *Brown, Bovey & Chen, 1997*; *Nazir, Sholla & Bashir, 2021*; *Ryan, Pascoe & Morse, 1997*; *Brown, 1995*; *Franklin & Flaschbart, 1998*; *Ward, Jones & Hopper, 1997*; *Hull, Neaves & Bedford-Roberts, 1997*).

Moreover, the advent of social networks has played a significant role in gathering user information, preferences, and behavioral patterns (*Sarker, 2018b*; *Sarker, 2018a*). Gadgets and applications have emerged as valuable sources of user data. For example, the SociaCircuit Platform enables the monitoring of social interactions, facilitating the adaptation of user preferences (*Chronis, Madan & Pentland, 2009*). Other studies have utilized data mining tools, sociometric badges, and mobile sensors to track user activities, predict job satisfaction, analyze employee interactions, and identify significant locations based on social activities (*Jung, 2009*; *Aly, Eskaf & Selim, 2017*; *Olguín et al., 2008*; *Eagle & Pentland, 2006*).

Despite efforts to integrate complex expert systems into the Android platform, certain limitations persist. For example, the book "Build Android-Based Smart Applications" describes using rule engines on Android (*Mukherjee, 2017*). However, these rule engines frequently need more critical characteristics like context awareness, resource efficiency, and dynamic context utilization with preferences. For example, some rule engines do not offer OR operators. In contrast, others require authoring rules in code or storing separate files for each rule, which can be inconvenient for bigger systems. Technical issues arose during the porting of rule engines. Due to Drools' memory-intensive nature, Eclipse encountered memory restrictions while converting files to Dalvik.

Similarly, Take required a Java compiler at runtime, while Jlisa experienced stack overflows on the Android platform. Furthermore, despite its compatibility concerns and high cost, the Jess rule engine posed challenges due to significant memory usage. Table 1 shows different rule engines available in the literature. Most of them are based on Rete algorithms, which are greedy algorithms that are not recommended for memory-constrained devices due to their memory-intensive nature.

Furthermore, *Uddin et al. (2016)* needs a systematic method for memory allocation instead of relying on fixed-size memory for execution, which may lead to problems

**Table 1 Comparison of various rule engines.**

| Name | Generic | Context-Aware | For resource bounded devices | Context removal | RETE based | Porting issue | Context reusability |
|---|---|---|---|---|---|---|---|
| Clips | Yes | No | No | No | Yes | Yes | NA |
| Drools | Yes | No | No | No | Yes | Yes | NA |
| FMES | No | No | No | No | No | NA | No |
| Jlisa | Yes | No | No | No | No | Yes | NA |
| JruleEngine | Yes | No | No | No | Yes | Yes | NA |
| JEOPS | Yes | No | No | No | Yes | Yes | NA |
| KBAM | No | Yes | No | No | No | NA | No |
| SociaCircuit | No | NA | No | No | NA | NA | Yes |
| Zilionis | Yes | No | No | No | Yes | Yes | NA |
| PSBM | Yes | Yes | Yes | Random | No | No | Yes |
| TermWare | Yes | No | No | No | Yes | No | No |

discussed in detail in *Ali et al. (2024)*. Notably, both *Uddin et al. (2016)* and *Uddin (2019)* need to incorporate a method for removing previously loaded rules from memory once it has reached capacity. Instead, rules are randomly removed, maybe including essential ones that will shortly be required for execution. This condition could send the system into an infinite loop if a vital rule needed for execution is inadvertently removed. The following subsection goes into preferences in greater detail.

## Preferences

The fundamental idea behind preferences is to select a subset of the rule base instead of the entire set. The preferences can have a significant impact on the overall performance of the system. This technique allows the inference engine to iterate through a subset of rules, enhancing the system's efficiency. This is particularly relevant for personalizing resource-constrained context-aware applications without modifying the primary rule base. Although preferences in multi-agent context-aware systems have received less attention (*Brandt, Chabin & Geist, 2015*), they hold importance in recommender applications (*Abbas, Zhang & Khan, 2015*), User Interface (UI) customization (*Loitsch et al., 2017*), rules for triggering context-awareness (*Moore & Pham, 2015*), personalizing notifications (*Auda et al., 2018; Mehrotra, Hendley & Musolesi, 2016*), and acting as tour guides (*Manotumruksa, Macdonald & Ounis, 2016*).

The basic approach in the methods above involves gathering user preferences and tailoring the user experience accordingly. Preferences were utilized for UI customization in terminals (*Loitsch et al., 2017*). The flow manager retrieves user preferences from a cloud-based server upon user login, enabling adjustments to the UI based on these choices and relevant information obtained from the user's device *via* USB or NFC channel. In *Alhamid et al. (2016)*, preferences were predicted from user-item selections, which can be an application software in the current context. User preferences can be computed based on the relationship between objects to uncover the underlying reasons for user behavior in specific situations. However, this approach may face a cold start problem due to limited initial information. Similarly, *Manotumruksa, Macdonald & Ounis (2016)* may encounter

the same challenge when visiting new places or in situations with limited information. In this article, the technique is context-aware, meaning that context can be derived from the words in the user's feedback remarks.

Location-based social networks (LBSN), such as Foursquare, offer context recommendations for places to visit. To make such a system context-aware, it should consider associated user information like the time of day and location. The authors employed word embedding to infer venue representation, user contextual preferences, and existing user preferences. *Zhu et al. (2014)* utilized context logs to mine a variety of popular context-aware preferences. However, these solutions require significant resources in terms of memory and computation, making them unsuitable for resource-constrained devices. To address this, *Uddin & Rakib (2018)* proposes a novel technique where users have control over their choices and are explicitly asked to indicate their interests in the knowledge base, enabling efficient resource utilization. While this approach seemed superior.

Compared to RETE algorithm-based solutions at the time, it lacked support for preferences once loaded. Over time, this approach accumulated significant overhead and consumed excessive memory, which posed a considerable challenge, especially considering the limited availability of memory resources. The current working memory updating technique is discussed in the next section.

## WORKING MEMORY SIZE ESTIMATION

Estimating the size of working memory proves advantageous in addressing context loss, particularly in critical contexts. In *Ali et al. (2024)*, we have achieved this using three distinct techniques: (i) Distinct Working Memory (DWM), (ii) Average of the Preference Sets (APS), and (iii) Smart Average of the Preference Sets (SAPS). While all three techniques perform effectively in their respective scenarios, SAPS emerges as the more viable option. It not only utilizes less memory compared to APS but also requires less time for estimating the working memory size in most scenarios. The APS technique lays the groundwork for SAPS to calculate the required working memory size, making it more practical for detailed discussion, as illustrated in Algorithm 1. The APS algorithm begins by initializing and measuring the time for performance assessment. It then determines the total number of preference sets and calculates the number of rules in each set. The algorithm computes the average number of regulations across all preference sets and creates a distinct rule set (DRS) if the calculated average is less than the number of rules in the preference set with the highest rule count (PSHR). The DRS is formed by including rules absent from the original rule set. Subsequently, the algorithm compares the number of regulations in the DRS with the computed average and sets the required working memory size (RWM) based on the outcome of this comparison. The time measurement concludes, and the algorithm returns the calculated RWM and the time taken for performance assessment. The conditions in the algorithm enable adaptive adjustments to memory requirements and processing time, making it suitable for scenarios where rule-based characteristics impact system performance.

---

**Algorithm 1:** Average of the Preference Sets (APS) (*Ali et al., 2024*)

**Input: RB:** Rule-base [**RC:** Rule Consequent, **RDC:** Rule Distinct Consequent, **RS:** Array to hold Rule Set, **DRS:** Array to hold distinct Rule Set, **NPS:** Number of Preference Set, **PS:** Preference Set, **APS:** Average of Preference sets, **PSHR:** Preference Set with highest Number of Rules]

**Output: RWM:** Required Working Memory Size, **TPM:** Time for Performance Measurement

**START**
**TPM_ start** to start measuring the time
**NPS** ← Check **RB** for the number of preference sets
**PS[i]** ← Check **RB** for the number of rules in each preference sets
**Average** ← Calculate the average/mean of the preference sets
**if (Average <PSHR), then**
Create **DRS**
**for** each Rule in **PSHR do**
**if RS** does not include **RC then**
Push rule to **DRS**
**end**
**end**
**end**
**if** Number of rules in **DRS >Average then**
**RWM** ← **DRS**
**end**
**else**
**RWM** ← **Average**
**End**
**TPM_ end**
**TPM = TPM_ start −TPM_ end**
Return **RWM, TPM**
**END**

---

## WORKING MEMORY UPDATE METHODOLOGIES

Working memory stores the currently available contexts, allowing for context-aware reasoning. Saving memory emerges as a critical problem during system design and implementation procedures prioritizing resource constraints. Limiting the size of working memory guarantees that it does not exceed its ability to hold contexts at any given time. However, contexts can be generated almost at every iteration. Therefore, it is critical to keep those that are more relevant for execution. The following section provides an overview of existing and proposed models for updating working memory.

## Current working memory update methodology

In the current implementation proposed by *Rakib & Haque (2014)*, the working memory is structured as a fixed-size container comprising static and dynamic components. The dynamic memory portion is limited in size, with each memory unit capable of storing a single context. Only facts stored in the dynamic memory are subject to being overwritten, a scenario triggered either by the agent's memory reaching capacity or the arrival of a contradictory context in the working memory, regardless of its current occupancy. Upon arriving at a newly derived context, it compares with existing contexts to detect any conflicts. If a conflict arises, the conflicting context is replaced with the newly derived one. If the working memory is full, the new context is added by overwriting a random existing context. Due to the finite size of the working memory, the system may enter into an infinite execution loop if a goal proves unachievable without a mechanism to forcibly halt the inference engine. Similarly, the context targeted for removal could be critical and potentially necessary for imminent execution.

The Rete algorithm working memory updating is only the introduction/loading of new rule(s)/context(s) to working memory, as the applications based on the Rete algorithm claim memory almost equal to the rules set provided; due to this, there might not be any chance that a rule does not have sufficient space in memory to be loaded.

## Proposed working memory update methodology

The memory size calculation has been proposed in our previous work *Ali et al. (2024)*. We can select one of the three methods (DWM, APS, and SAPS) for memory calculation. After calculating the memory, the currently proposed technique is the final step toward memory-efficient utilization. Algorithm 1 shows the technique in detail.

This algorithm has three major parts: (i) First, the system will check whether the context(s) is loading for the first time through WM Flag. If it is false, there is no data, and the data is loading to the memory for the first time. So, the system will load the data directly without checking the other attributes. (ii) If the context(s) are already there and there is a new incoming context, the system will first check the preference. If the preference matches, then it means that the new context belongs to the same preference set. After this, the system will check the context with the existing contexts. The system will load the context into the memory if its consequent part is distinct. Conversely, the system will overwrite the context over the existing context, which will have the same consequence. (iii) If the preference does not match the existing one in memory, the system will then check if the preference matches any consequent part. If yes, the rule will be appended in memory in the next available free slot. In case of no, it is time to change the context(s) loaded in the memory as the incoming context shows the change of preference set. So, the system will remove all the rules of the old preference set, update the preference in memory, and start loading the rule(s) of the new preference set.

Algorithm 2 completes the framework for which the memory calculation model was introduced in our previous article, *Ali et al. (2024)*. This algorithm minimizes the load over the memory by removing the context(s)that are no longer required, as the preference set has been changed. This algorithm overcomes the issue of critical context(s) removal, as the

---

**Algorithm 2:** Proposed Working Memory Update Algorithm (WMUA)

---

**Input:** Fact(s), Context(s)
**START**
load **Facts**
check **WM_Flag**
**If** (**WM_** *Flag* $==$ *False*) **then**
set **WM_Flag** $\leftarrow$ **True**
load **Preference**
load **Context** to **WM**
**end**
**else**
check **Preference**
**if** (**Preference Matches**) **then**
check rule for **Distinct_Consequent** Part
**if** (**Consequent is Distinct**) **then**
append the rule in **WM**
**end**
**else**
**Overwrite** the rule(Over the same consequent rule in WM)
**end**
**else if** (**Preference Matches any Consequent Part**) **then**
**Overwrite** the rule in **WM**
**end**
**else**
clear the **WM**(Except the rule having "**Tell**" part)
load **Preference**
load **Context in WM**
**end**
**end**
**END**

---

system now has sufficient memory to load all the required contexts, and there may not be a need to remove context(s) randomly due to lack of space in memory.

Figure 1 (Step 1) illustrates the preprocessing part introduced by the authors of *Uddin (2019)*, where the rule set creation and preference set were introduced and followed in the same manner in this article. Figure 1 (Step 2) corresponds to our previous work published in *Ali et al. (2024)*, where three algorithms, DWM, APS, and SAPS, have been introduced for the calculation of required working memory; these algorithms are also followed in the same context as they are working in the article which introduced it. Figure 1 (Step 3 WMUA) outlines the proposed workflow of this article that removes the preference set, which is no longer required for processing. In the next section, two different scenarios have been discussed to validate the proposed work.

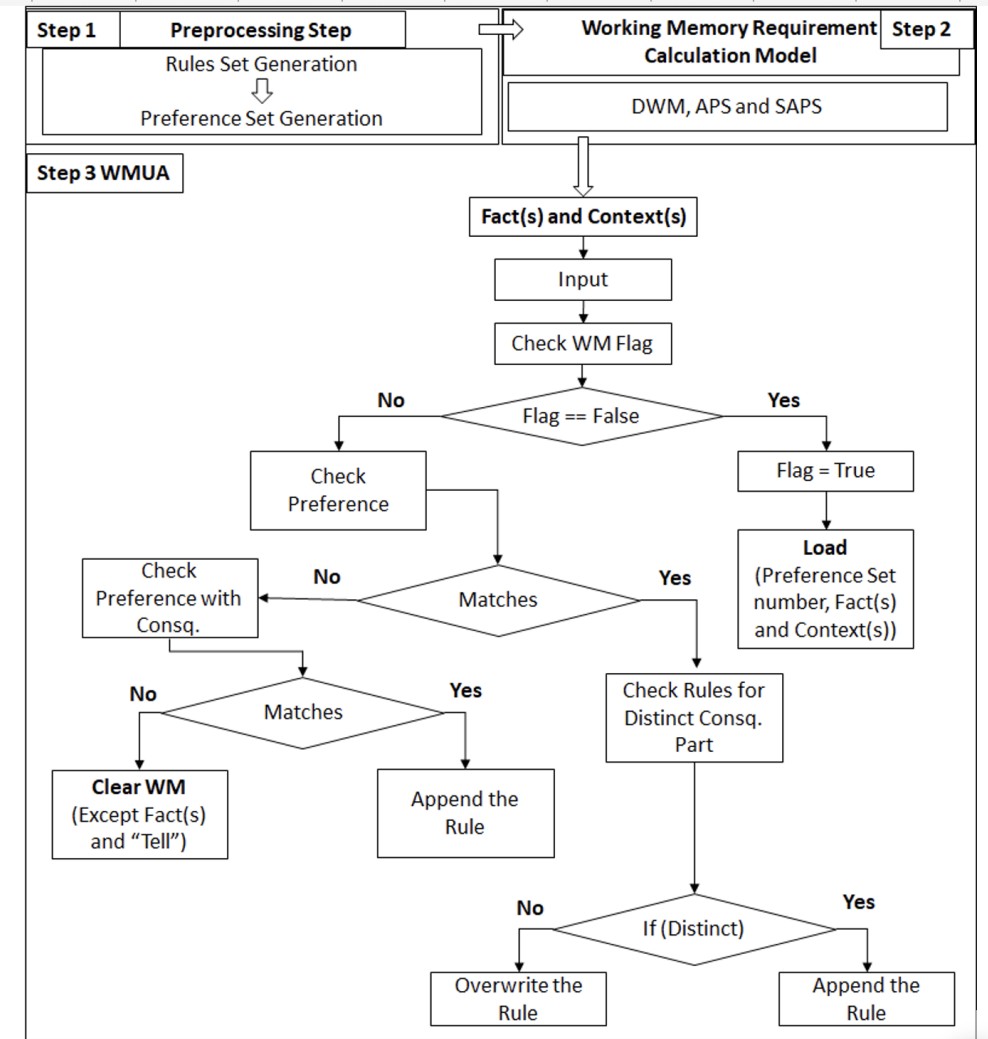

**Figure 1  Integrated framework with latest proposed work.**

## SCENARIOS

Here are the two scenarios employed to assess the model's performance. The implementation has been completed, and results have been generated using a Raspberry Pi 3 Model B with 1 GB of RAM.

### Scenario 1 (patient care system)

The presented scenario is documented in *Uddin (2019)*, encompasses 85 rules, and revolves around a patient care system. The ontology outlining the structure of this scenario is visually depicted in Fig. 2. While varied and extensive sets of steps for annotation and preference sets exist within this scenario, our focus in this discussion is primarily on memory usage. Notably, the creation and execution of preference sets will remain consistent.

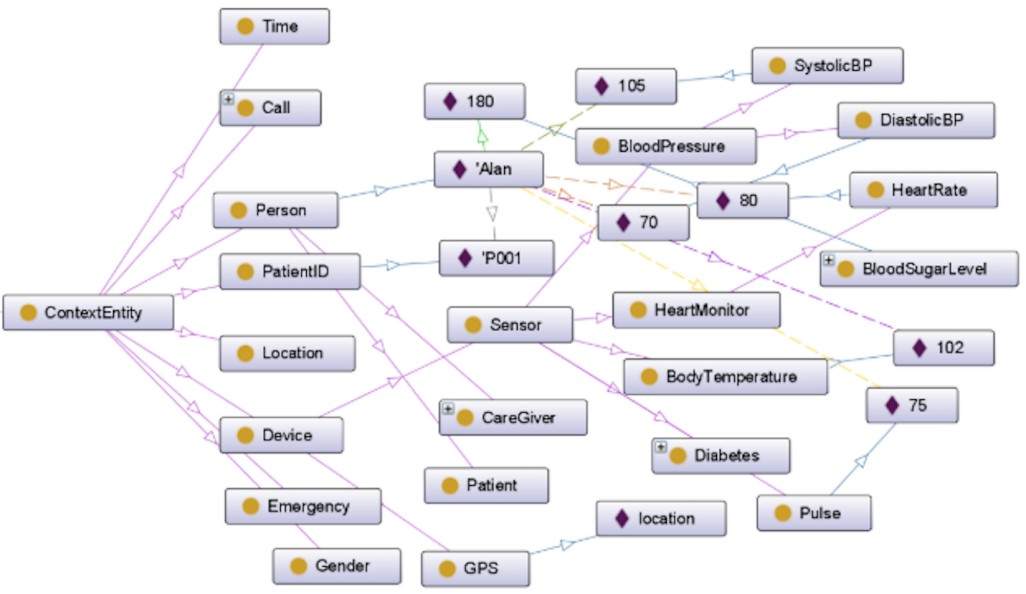

**Figure 2  Patient care ontology.**

```
Rules:
{"Agent Name and Facts": "Patient Care Agent ( 1)", "Condition": "PatientID(?pid), Person(?p), hasPatientID(?p, ?pid)",
 "Consequent": "Patient(?p)","Preference": "."},
{"Agent Name and Facts": "PatientID(01), Person(Alan), hasPatientID(Alan,01)", "Condition": "Patient(?p),
 hasFever(?p, Hyperpyrexia)", "Consequent": "hasSituation(?p, Emergency)", "Preference": "."},
{"Condition": "Patient(?p), hasBloodPressure(?p, High), hasDBCategory(?p, Low),hasFever(?p, High), hasPulseRate(?p,High)",
 "Consequent": "hasSituation(?p, Emergency)", "Preference": "GPSLocation(Home)"},
{"Condition": "Patient(?p), hasBloodPressure(?p, Low), hasDBCategory(?p, Low), hasFever(?p, High), hasPulseRate(?p,Low)",
 "Consequent": "hasSituation(?p, Emergency)", "Preference": "GPSLocation(Home)"},
{"Condition": "Patient(?p), hasBloodPressure(?p, High), hasDBCategory(?p, High), hasFever(?p, High), hasPulseRate(?p,High)",
 "Consequent": "hasSituation(?p, Emergency)", "Preference": ""GPSLocation(Office)""},
{"Condition": "Patient(?p), hasPulseRate(?p, Abnormal)", "Consequent": "hasSituation(?p, Emergency)",
 "Preference": ""GPSLocation(Home)""},
{"Condition": "Patient(?p), hasDBCategory(?p, Normal)", "Consequent": "hasNotSituation(?p, Emergency)",
 "Preference": "GPSLocation(Hospital)"},
{"Condition": "Nurse(?nurse), Patient(?p), hasBloodPressure(?p, Low)", "Consequent": "isCaredBy(?p, ?nurse)",
 "Preference": "GPSLocation(Hospital)"},
```

**Figure 3  A snapshot of some of the patient care rules.**

The development of the rule set yielded seven preference sets, exemplified by a sample of rules illustrated in Fig. 3. The rules set and preference set were generated following the methodology introduced by *Uddin (2019)*. This snapshot provides a glimpse into the complexity and richness of the rule-based system designed for the patient care scenario, as detailed in the comprehensive case study. For memory calculation, APS was employed, as SAPS and DWM do not align well with the requirements of this scenario (*Ali et al., 2024*). Since the standard deviation exceeded 2, it is advised not to adhere to SAPS, as recommended by the authors. The proposed algorithm executed memory updating.

The RETE algorithm poses a challenge due to its memory-intensive nature, loading almost all 85 rules into memory without considering preference sets (on Raspberry PI 3B, it consumed 405 bytes). In contrast, the preference-sets-based method, as observed in our case, demands a minimum space for 40 rules with a memory requirement of

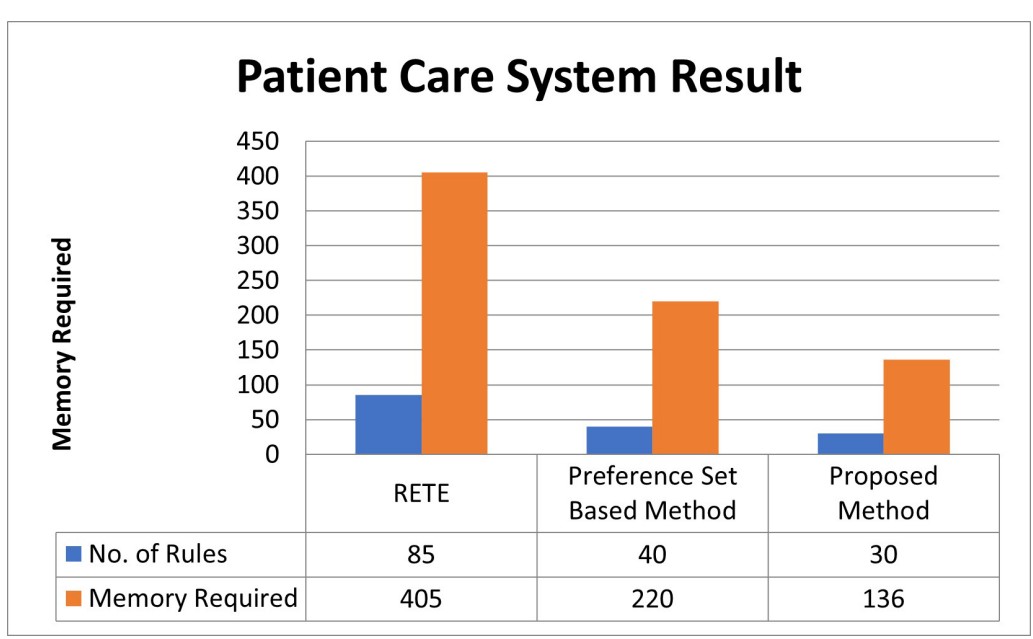

**Figure 4** **Patient care system result.**

220 bytes. Notably, the absence of a memory requirement calculation technique in the preference-sets-based method leads to random memory allocation, an issue effectively addressed in our prior work *Ali et al. (2024)*.

While the preference-sets-based method exhibits a nearly 59% reduction in memory requirements in this instance, it encounters difficulties when the available memory lacks space for incoming rules or contexts. This method lacks a mechanism to remove preference sets no longer in use. Consequently, when a new context emerges, the method randomly removes existing contexts to accommodate the incoming one. This random removal process introduces the risk of discarding critical contexts, potentially leading to unforeseen issues or, in some cases, causing an infinite loop.

In our prior article, *Ali et al. (2024)*, we introduced three distinct methods, DWM, APS, and SAPS, enabling the calculation of memory requirements. For the same set of rules, these methods yield memory requirements of 45 rules (236 bytes), 30 rules (136 bytes), and 30 rules (136 bytes), respectively (APS employed here). Integrating these methods with the currently proposed one forms a comprehensive framework. This framework excels by eliminating preference sets no longer needed for processing, ensuring a systematic approach. Consequently, this technique eradicates the necessity for random removal of contexts, enabling the system to operate seamlessly. Notably, it exhibits a remarkable efficiency with a percentage difference of 22.22% compared to the Preference-Sets-Based-Method and 78.82% compared to RETE-Based-Algorithms, contributing to a significantly improved system performance described in Fig. 4.

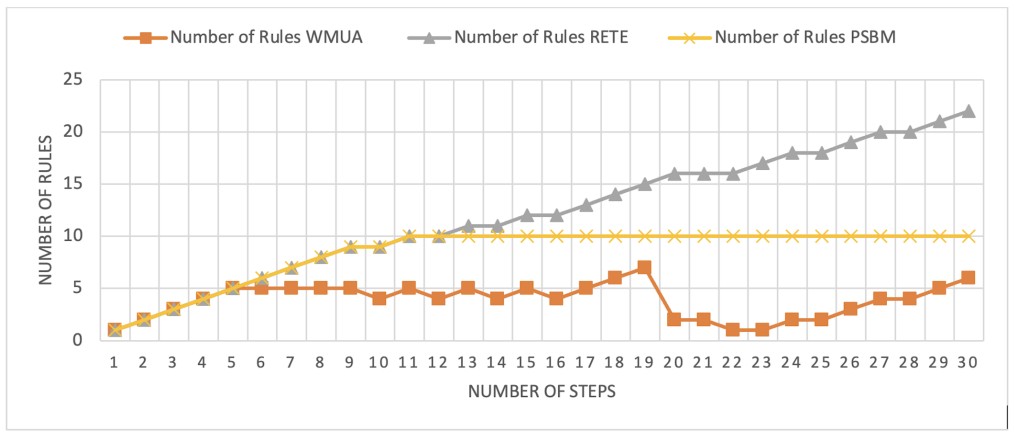

**Figure 5** Comparative results for scenario 1 (PSBM with 10 rules space).

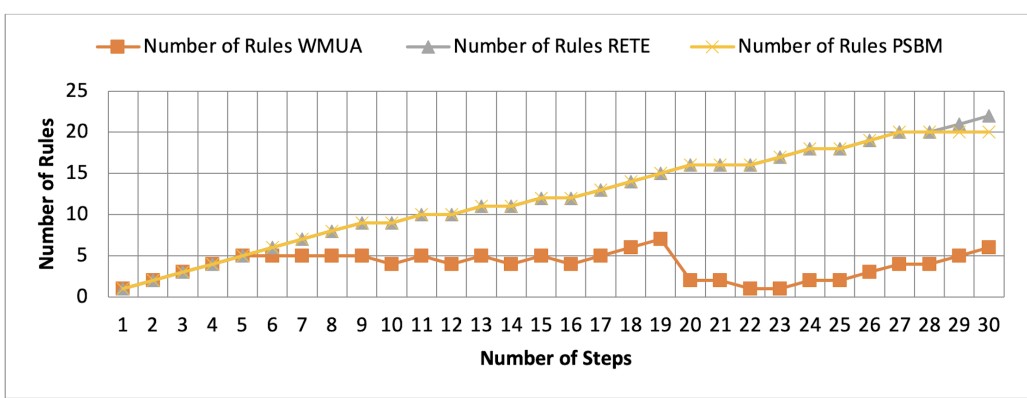

**Figure 6** Comparative results for scenario 1 (PSBM with 20 rules space).

## Working memory updating results for scenario 1

Table 2 presents working memory update steps that were utilized for the comparative analysis of three algorithms, WMUA, RETE, and PSBM, based on the number of rules involved at different steps of their execution. Some of the steps can run in parallel in the patient care scenario. However, to better understand the results, we implemented it sequentially in Table 2. In total, there are 30 steps in the scenario. The Rete and the proposed WMUA algorithm were implemented per the algorithm recommendation. However, as the PSBM algorithm claims memory randomly, we implement it with different memory claims three times for better understanding and comparison. The memory required first for ten rules, second for 20 rules, and third for 40 rules, as shown in Figs. 5, 6, and 7, respectively. The figures show that the more we increase the memory size, the more PSBM behaves like the Rete algorithm regarding memory consumption. In Fig. 7, both are exact matches.

Similarly, suppose we chose less memory for the PSBM algorithm. In that case, rules need to be randomly removed from the memory to make room for incoming rules, which

**Table 2  Working memory update steps for scenario 1.**

| Steps number | Agent number | Rule number | Preference | Consequent |
|---|---|---|---|---|
| 1 | 1 | 1 | . | Patient(?p) |
| 2 | 2 | 45 | . | hasBPLevel(?p, High) |
| 3 | 3 | 56 | . | hasDBCat.(?p, High) |
| 4 | 4 | 65 | . | hasFever(?p, High) |
| 5 | 5 | 70 | . | hasPulseRate(?p, High) |
| 6 | 2 | 51 | hasBPLevel(?p, High) | Tell( 2, 1, hasBPLevel(?p, High)) |
| 7 | 3 | 60 | hasDBCat.(?p, High) | Tell( 3, 1, hasDBCat.(?p, High)) |
| 8 | 4 | 67 | hasFever(Mike, High) | Tell( 4, 1, hasFever(?p, High)) |
| 9 | 5 | 77 | hasPulseRate(?p, High) | Tell( 5, 1, hasPulseRate(?p, High)) |
| 10 | 1 | 51 | Copy | Copy |
| 11 | 1 | 36 | Patient(Mike) | hasBPLevel(?p, High) |
| 12 | 1 | 60 | Copy | Copy |
| 13 | 1 | 29 | Patient(Mike) | hasDBCat.(?p, High) |
| 14 | 1 | 67 | Copy | Copy |
| 15 | 1 | 30 | Patient(Mike) | hasFever(?p, High) |
| 16 | 1 | 77 | Copy | Copy |
| 17 | 1 | 33 | Patient(Mike) | hasPulseRate(?p, High) |
| 18 | 1 | 5 | Patient(Mike) | hasSituation(?p, Critical) |
| 19 | 1 | 42 | Patient(Mike) | Tell( 1, 6, hasSituation(?p, Critical)) |
| 20 | 1 | 44 | GPSLoc(Hospital) | hasBPLevel(?p, Hypotention) |
| 21 | 1 | 42 | Copy | Copy |
| 22 | 6 | idle | idle | idle |
| 23 | 6 | 79 | . | hasSituation(?p, Critical) |
| 24 | 6 | 81 | . | hasGPSLoc(?p, ?loc) |
| 25 | 7 | 81 | Copy | Copy |
| 26 | 7 | 84 | . | hasGPSLoc(?p, ?loc) |
| 27 | 7 | 85 | . | Tell( 7, 6, hasGPSLoc(?p, ?loc)) |
| 28 | 6 | 85 | Copy | Copy |
| 29 | 6 | 81 | . | hasGPSLoc(?p, ?loc) |
| 30 | 6 | 82 | . | isDiagnosedBy(?p, ?physc) |

leads to other issues that have been thoroughly discussed in *Ali et al. (2024)*. WMUA consistently demonstrates the lowest number of rules among the three algorithms across most steps.

This suggests that WMUA is adept at managing and processing rules with almost no redundancy, making it an attractive choice for scenarios where computational resources are limited or speed is critical. The average percentage difference in Rete and WMUA algorithm results is 103.49%, meaning WMUA needs almost half of the DWM compared to Rete. Similarly, WMUA also outperformed the PSBM algorithm with a minimum average percentage difference of 74.45%. However, when the memory size for PSBM increases, the difference also increases. These results make the WMUA algorithm better for memory and processing-constrained devices.

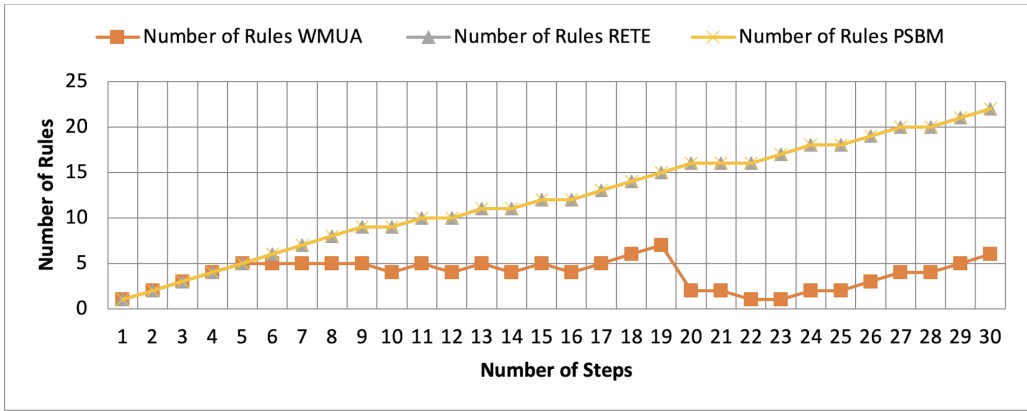

**Figure 7** Comparative results for scenario 1 (PSBM with 40 rules space).

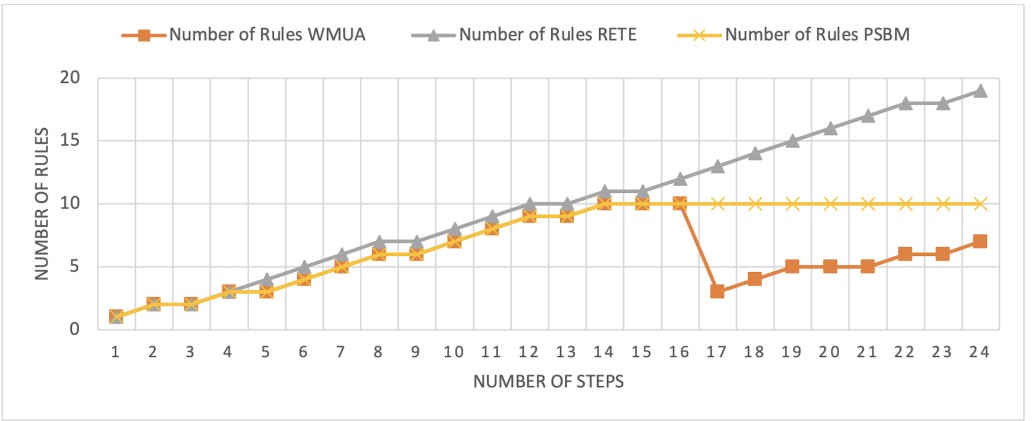

**Figure 8** Comparative results for scenario 2 (PSBM with 10 rules space).

## Scenario 2 (smart home)

The smart home scenario has also been published in *Uddin (2019)*. The total number of rules in this scenario is 32, and the number of preferences sets in this rule set is 2. The ontology outlined in Fig. 8 shows the structure of the scenario. Table 3 shows the steps involved in scenario 24 in number. Here, we also consider the sequential fashion for simplicity and a better understanding of the scenario's results. These 24 steps were utilized to analyze the performance of the three algorithms.

## Working memory updates results for scenario 2

The Rete and WMUA algorithms were implemented as per the recommendation of the algorithms in both the results shown in Figs. 8 and 9. However, for the PSBM algorithm in the first experiment, the memory was fixed for 10 rules and in the second for 20 rules, respectively. In these results, PSBM and WMUA perform similarly till step number 16; after this step, there is a preference change, and the WMUA algorithm removes the rules

**Table 3  Working memory update steps for scenario 2.**

| Steps number | Agent | Rule # | Preference | Consequent |
|---|---|---|---|---|
| 1 | 8 | 86 | GPSLoc(home) | isAuthorizedPerson(?p, Yes) |
| 2 | 8 | 87 | GPSLoc(home) | Tell( 8, 9, isAuthenticPerson(?p, Yes)) |
| 3 | 9 | 87 | copy | copy |
| 4 | 8 | 88 | GPSLoc(home) | Tell( 8, 15, isAuthenticPerson(?p, Yes)) |
| 5 | 9 | 94 | GPSLoc(home) | isAuthenticPerson(?p, Yes) |
| 6 | 9 | 92 | GPSLoc(home) | hasOccupancy(?p, Yes) |
| 7 | 9 | 89 | GPSLoc(home) | Tell( 9, 10, hasOccupancy(?p,Yes)) |
| 8 | 9 | 90 | GPSLoc(home) | Tell( 9, 11, hasOccupancy(?p,Yes)) |
| 9 | 10 | 90 | copy | copy |
| 10 | 9 | 91 | GPSLoc(home) | Tell( 9, 12, hasOccupancy(?p,Yes)) |
| 11 | 10 | 100 | GPSLoc(home) | hasAirconStatus(?room, On) |
| 12 | 9 | 93 | GPSLoc(home) | Tell( 9, 10, isAvailableAt(?p,?room)) |
| 13 | 11 | 90 | copy | |
| 14 | 8 | 95 | GPSLoc(home) | Tell( 9, 11, isAvailableAt(?p,?room)) |
| 15 | 10 | 95 | copy | copy |
| 16 | 11 | 102 | GPSLoc(home) | hasOccupancy(?p, Yes) |
| 17 | 10 | 96 | isAuthenticPerson(Ali,yes) | isAvailableAt(?p, ?room) |
| 18 | 11 | 95 | GPSLoc(home) | Tell( 9, 11, isAvailableAt(?p,?room)) |
| 19 | 13 | 113 | GPSLoc(home) | hasTemp(?t, Hot) |
| 20 | 10 | 99 | isAuthenticPerson(Ali,yes) | hasLightStatus(?p, On) |
| 21 | 11 | 105 | GPSLoc(home) | isAvailableAt(?p, ?room) |
| 22 | 13 | 112 | GPSLoc(home) | Tell( 13, 11, hasTemp(?t,Hot)) |
| 23 | 11 | 112 | copy | copy |
| 24 | 11 | 100 | GPSLoc(home) | hasAirconStatus(?room, On) |

and clears space for upcoming rules. The PSBM memory limit in the first experiment is 10, and it reaches its limit on step number 14. After that, the algorithm needs to remove a rule randomly for each new upcoming rule, which may lead to different problems discussed in our previous article (*Ali et al., 2024*) in detail. In experiment number 2, shown in Fig. 9, it did not reach its limit (that is 20 rules at most) and continuously moved up word by introducing new rules to memory. The Rete algorithm showed its memory-intensive nature from step number 4 and onward in both results shown in Figs. 8 and 9.

The average percentage difference here in this scenario also follows a similar trend. That is the average percentage difference between Rete and WMUA in this scenario is 54.34%. Similarly, the minimum difference between PSBM and WMUA calculated for the result shown in Fig. 8 is 25%, and it will increase for the result shown in Fig. 9 which becomes 34.57%. Here, in this scenario, the WMUA also outperformed both algorithms.

## CONCLUSION

This study introduces a systematic and innovative solution to tackle the dynamic working memory challenge within devices constrained by limited memory. Notably, the absence

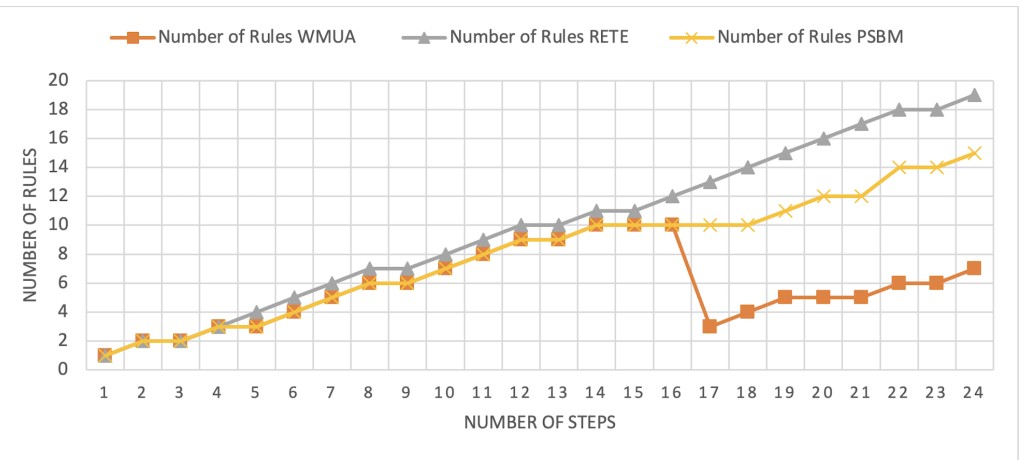

**Figure 9** Comparative results for scenario 2 (PSBM with 20 rules space).

of a methodical approach for removing context(s) from working memory had been a notable gap in prior research. Existing methods either retained loaded contexts without discernment or resorted to random removal, introducing critical issues as meticulously identified within the article.

The proposed initiative seeks to revolutionize the landscape by offering a systematic methodology for efficiently managing working memory in resource-constrained devices. By addressing the previous shortcomings, this solution opens avenues for constructing more intricate and intelligent systems tailored to operate seamlessly on devices with restricted resources. The significance lies in not only optimizing memory utilization but also in providing a reliable mechanism for the controlled removal of contexts, thereby enhancing the overall performance and reliability of intelligent systems on devices with constrained resources. This marks a crucial step forward in the pursuit of creating efficient and sophisticated systems capable of maximizing functionality even within resource limitations.

### Funding

This research work was funded by Institutional Fund Projects under grant no. (IF-PIP: 211-611-1443). Technical and financial support was provided by the Ministry of Education and King Abdulaziz University, DSR, Jeddah, Saudi Arabia. The funders had no role in study design, data collection and analysis, decision to publish, or preparation of the manuscript.

### Grant Disclosures

The following grant information was disclosed by the authors:
Institutional Fund Projects: IF-PIP: 211-611-1443.
The Ministry of Education and King Abdulaziz University, DSR, Jeddah, Saudi Arabia.

## Competing Interests

Osama Sohaib is an Academic Editor for PeerJ Computer Science.

## Author Contributions

- Mumtaz Ali conceived and designed the experiments, performed the experiments, analyzed the data, performed the computation work, prepared figures and/or tables, and approved the final draft.
- Muhammad Arshad conceived and designed the experiments, performed the experiments, analyzed the data, authored or reviewed drafts of the article, and approved the final draft.
- Ijaz Uddin conceived and designed the experiments, analyzed the data, authored or reviewed drafts of the article, and approved the final draft.
- Muhammad Binsawad analyzed the data, authored or reviewed drafts of the article, and approved the final draft.
- Abdullah Bin Sawad analyzed the data, authored or reviewed drafts of the article, and approved the final draft.
- Osama Sohaib analyzed the data, authored or reviewed drafts of the article, and approved the final draft.

## Data Availability

The code and dataset are available in the Supplementary Files.

## Supplemental Information

Supplemental information for this article can be found online at http://dx.doi.org/10.7717/peerj-cs.2129#supplemental-information.

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
