# Peer review of "Efficient context-aware computing: a systematic model for dynamic working memory updates in context-aware computing"

_PeerJ Computer Science, doi:10.7717/peerj-cs.2129_

## Round 0.1 · original submission · Major Revisions

Based on the reviewer comments, the paper must be accurately revised. Please give high importance to Reviewer 3.

**Language Note:** The review process has identified that the English language must be improved. PeerJ can provide language editing services - please contact us at copyediting@peerj.com for pricing (be sure to provide your manuscript number and title). Alternatively, you should make your own arrangements to improve the language quality and provide details in your response letter. – PeerJ Staff

Reviewer 1 ·

Basic reporting

The manuscript entitled “Efficient context-aware computing: A systematic model for dynamic working memory updates in context-aware computing” has been investigated in detail. The paper discusses the challenges and solutions regarding context-aware computing in resource-constrained devices, focusing on efficient management of working memory. It highlights the need for systematic content removal to optimize memory utilization. However, the paper lacks clarity in defining terms, providing concrete examples, and detailing the proposed methodology for memory management. Clearer explanations and specific examples are needed to strengthen the paper's contribution to context-aware computing. There are some points that need further clarification and improvement:
1) The readability and presentation of the study should be further improved. The paper suffers from language problems.
2) The introduction lacks a clear and concise statement of the problem that the paper aims to address. It vaguely discusses the landscape of ubiquitous computing without clearly delineating the specific issue being tackled.
3) The paper mentions "smart gadgets" and "smart devices" without providing concrete examples or defining the scope of these terms. Clarification on the types of devices and their functionalities would enhance the reader's understanding.

Experimental design

While the paper discusses context-aware computing and smart spaces, it does not adequately define these terms or provide examples to illustrate their relevance in everyday scenarios. This lack of contextualization diminishes the clarity of the discussion.

The paper introduces rule-based reasoning as an implementation approach for context-aware systems but does not sufficiently explain its relevance or effectiveness in comparison to other approaches. Providing a comparative analysis would strengthen the argumentation.

Validity of the findings

The discussion on working memory capacity and its efficient control is promising, but the paper does not delve into specific methodologies or algorithms proposed to address this challenge. Without concrete details, the proposed solution remains unclear.

“Discussion” section should be added in a more highlighting, argumentative way. The author should analysis the reason why the tested results is achieved.

The paper mentions introducing a systematic method for content removal to address the dynamic working memory challenge, but it lacks elaboration on the specifics of this method. Readers require more information on how content removal will be implemented and its expected impact on memory utilization.

Additional comments

The manuscript entitled “Efficient context-aware computing: A systematic model for dynamic working memory updates in context-aware computing” has been investigated in detail. The paper discusses the challenges and solutions regarding context-aware computing in resource-constrained devices, focusing on efficient management of working memory. It highlights the need for systematic content removal to optimize memory utilization. However, the paper lacks clarity in defining terms, providing concrete examples, and detailing the proposed methodology for memory management. Clearer explanations and specific examples are needed to strengthen the paper's contribution to context-aware computing. There are some points that need further clarification and improvement:
1) The readability and presentation of the study should be further improved. The paper suffers from language problems.
2) The introduction lacks a clear and concise statement of the problem that the paper aims to address. It vaguely discusses the landscape of ubiquitous computing without clearly delineating the specific issue being tackled.
3) The paper mentions "smart gadgets" and "smart devices" without providing concrete examples or defining the scope of these terms. Clarification on the types of devices and their functionalities would enhance the reader's understanding.
4) While the paper discusses context-aware computing and smart spaces, it does not adequately define these terms or provide examples to illustrate their relevance in everyday scenarios. This lack of contextualization diminishes the clarity of the discussion.
5) The paper introduces rule-based reasoning as an implementation approach for context-aware systems but does not sufficiently explain its relevance or effectiveness in comparison to other approaches. Providing a comparative analysis would strengthen the argumentation.
6) The discussion on working memory capacity and its efficient control is promising, but the paper does not delve into specific methodologies or algorithms proposed to address this challenge. Without concrete details, the proposed solution remains unclear.
7) The paper mentions introducing a systematic method for content removal to address the dynamic working memory challenge, but it lacks elaboration on the specifics of this method. Readers require more information on how content removal will be implemented and its expected impact on memory utilization.
8) Overall, the paper lacks depth in its discussion and fails to provide sufficient details on the proposed solution and its potential implications. It needs to provide clearer explanations, concrete examples, and a more detailed methodology to enhance its credibility and relevance in the field of context-aware computing.
9) “Discussion” section should be added in a more highlighting, argumentative way. The author should analysis the reason why the tested results is achieved.
10) It will be helpful to the readers if some discussions about insight of the main results are added as Remarks.
This study may be proposed for publication if it is addressed in the specified problems.

Reviewer 2 ·

Basic reporting

no comment

Experimental design

no comment

Validity of the findings

no comment

Additional comments

This paper discusses the interesting and important issue of solving the problem of dynamic working memory in memory-limited devices by implementing a systematic method of content removal to improve the creation of intelligent systems. However, due to the fact that the basis of this research is covered in previous studies, it is not possible to assess the quality and consistency at this stage. The interrelationships of different approaches and the assessment of their effectiveness remain unclear. Also, the graphic materials provided may not be entirely appropriate. In general, the article needs to be thoroughly revised to create a unified order of coverage of the material, which will provide readers with a clear research design supported by the results.

Reviewer 3 ·

Basic reporting

The manuscript entitled “Efficient context-aware computing: A systematic model for dynamic working memory updates in context-aware computing” has been investigated in detail. The paper presents a method for managing dynamic working memory in resource-constrained devices for context-aware computing. However, it lacks novelty, comprehensive analysis of existing solutions, technical details, justification, and empirical evaluation. Significant revisions and additional experiments are needed to enhance its quality and credibility

Experimental design

The paper lacks empirical evaluation of the proposed method. Without experimental results or comparative analyses with existing approaches, it is challenging to assess the effectiveness and performance of the proposed solution.

Validity of the findings

1) The paper lacks novelty as it primarily addresses a well-known challenge in context-aware computing - efficient memory management in resource-constrained devices. The introduction of a systematic method for content removal does not sufficiently contribute to the advancement of the field.

2) While the paper discusses the challenge of dynamic working memory in resource-constrained devices, it fails to provide a comprehensive review of existing solutions and their limitations. Without a thorough analysis of the current state-of-the-art methods, the significance of the proposed method is questionable.

3) The paper lacks in-depth technical details regarding the proposed method for content removal. It does not provide clear algorithms or methodologies for implementing the proposed approach, making it difficult for readers to understand the feasibility and effectiveness of the proposed solution.

4) The paper does not adequately justify the need for a new method for content removal in the context of dynamic working memory. It does not provide evidence or examples illustrating the limitations of existing approaches or the specific scenarios where the proposed method would outperform them.

Additional comments

Overall, the paper does not meet the standards for publication due to its lack of novelty, limited scope, weak technical details, insufficient justification, and lack of empirical evaluation. Significant revisions and additional experimental validation are necessary to improve the quality and credibility of the research.

---

## Round 0.2 · accepted · Accept

Based on the concers of the reviewer 2, the authors clarified the effectiveness of the interrelationships of different approaches. The images were also improved to support the results. The other two reviewers given a positive feedback. In this case, the manuscript can be accepted.

Reviewer 1 ·

Basic reporting

My comments have been addressed. It is acceptable in the present form.

Experimental design

My comments have been addressed. It is acceptable in the present form.

Validity of the findings

My comments have been addressed. It is acceptable in the present form.

Reviewer 3 ·

Basic reporting

The study can be accepted for publication.

Experimental design

The study can be accepted for publication.

Validity of the findings

The study can be accepted for publication.